# Biofluid Biomarkers in the Prognosis of Amyotrophic Lateral Sclerosis: Recent Developments and Therapeutic Applications

**DOI:** 10.3390/cells12081180

**Published:** 2023-04-18

**Authors:** Daniel Sanchez-Tejerina, Arnau Llaurado, Javier Sotoca, Veronica Lopez-Diego, Jose M. Vidal Taboada, Maria Salvado, Raul Juntas-Morales

**Affiliations:** 1Neuromuscular Diseases Unit, Neurology Department, Vall d’Hebron Hospital Universitari, Vall d’Hebron Barcelona Hospital Campus, 08035 Barcelona, Spain; 2Peripheral Nervous System Group, Vall d’Hebron Research Institut (VHIR), Vall d’Hebron Barcelona Hospital Campus, 08035 Barcelona, Spain; 3European Reference Network on Rare Neuromuscular Diseases (ERN EURO-NMD), Vall d’Hebron Barcelona Hospital Campus, 08035 Barcelona, Spain; 4Medicine Department, Universitat Autónoma de Barcelona, 08035 Barcelon, Spain

**Keywords:** amyotrophic lateral sclerosis (ALS), biomarker, prognosis, pharmacodynamic biomarker, neurofilament light (NfL) protein, neuroinflammation, genetics

## Abstract

Amyotrophic lateral sclerosis is a neurodegenerative disease characterized by the degeneration of motor neurons for which effective therapies are lacking. One of the most explored areas of research in ALS is the discovery and validation of biomarkers that can be applied to clinical practice and incorporated into the development of innovative therapies. The study of biomarkers requires an adequate theoretical and operational framework, highlighting the “fit-for-purpose” concept and distinguishing different types of biomarkers based on common terminology. In this review, we aim to discuss the current status of fluid-based prognostic and predictive biomarkers in ALS, with particular emphasis on those that are the most promising ones for clinical trial design and routine clinical practice. Neurofilaments in cerebrospinal fluid and blood are the main prognostic and pharmacodynamic biomarkers. Furthermore, several candidates exist covering various pathological aspects of the disease, such as immune, metabolic and muscle damage markers. Urine has been studied less often and should be explored for its possible advantages. New advances in the knowledge of cryptic exons introduce the possibility of discovering new biomarkers. Collaborative efforts, prospective studies and standardized procedures are needed to validate candidate biomarkers. A combined biomarkers panel can provide a more detailed disease status.

## 1. Introduction

Amyotrophic Lateral Sclerosis (ALS) is the most common adul-onset motor neuron disease. Currently defined as a neurodegenerative disease, ALS patients develop a progressive, although variable, degeneration of both the upper and lower motor neurons in the motor cortex, brainstem and spinal cord anterior horn [1,2]. It is a globally distributed condition with an incidence of 0.6–3.8 per 100,000 person per year and a prevalence between 4.1 and 8.4 per 100,000 persons [3]. The progression of the disease and the loss of motor neurons leads to the denervation of voluntary muscles, amyotrophy and spreading muscle weakness. ALS is considered to be a devastating disease, resulting in extensive paralysis, and eventually, death, usually due to respiratory muscle dysfunction [1,2]. Although life expectancy from diagnosis averages 2–5 years, the clinical heterogeneity of the disease manifests itself not only in the different patterns of motor involvement, but also in the rate of progression, which may include early death within a few months or survival of more than ten years [3,4].

In addition to this heterogeneity in the clinical manifestations of the disease, research carried out in the past decades has revealed remarkable molecular complexity in the pathophysiology of ALS, with a cascade of multiple pathways involved, as well as an increasingly important complex genetic substrate [5,6]. Despite advances in our knowledge, repeated efforts to translate it into effective therapy have been unfruitful. There are several possible causes for these failures, but one of the factors that has been consistently highlighted is the need for robust and reliable biomarkers in ALS [7,8]. The statements from the TRICALS consortium (Treatment Research Initiative to Cure ALS) and the revised Airlie House ALS Clinical Trials Consensus Guidelines are remarkable examples of this consensus within the international ALS community [9,10]. Both entities represent the common interests of ALS researchers, clinicians, patients and industry, and they concur that the existence of validated biomarkers is one of the necessities for research in ALS.

In this review, we aim to discuss the current state of the evidence of prognostic and predictive biomarkers of ALS in biological fluids, with a particular emphasis on those that are expected to be incorporated into the development of new therapies in clinical trials and routine clinical practice. However, while the path to biomarker discovery and validation is undoubtedly complex, and its discussion is beyond the scope of this review, we would like to begin by briefly reviewing the conceptual and operational framework of biomarkers in the field of ALS. The first section focuses on biomarker categorization based on targeted use, and the second section discusses the differences between available sources of fluid-based biomarkers.

## 2. Biomarkers in ALS: Categories, Conceptualization and Rational Application

The commonly accepted definition of the term biomarker is a “characteristic that is objectively measured and evaluated as an indicator of normal biological processes, pathologic conditions, or biological response to a therapeutic intervention [9,11]”. However, the concept of biomarker needs to be nuanced, considering that different types of biomarkers are recognised depending on their intended use. The National Institutes of Health (NIH) and the Food and Drug Administration (FDA) in the United States have recently launched an initiative called the BEST (Biomarkers, EndpointS and other Tools) Resource aimed at reaching a consensus on the terminology used in translational research [12]. In this theoretical framework, a key aspect is that the differences between the classes of biomarkers should be governed by the “fit for purpose” concept. This means that the methodology of description and validation, and especially their application, should be adjusted for the required use. In the context of ALS, the most important biomarker categories are the following ones (a more detailed description and illustrative examples are given in Table 1).

A diagnostic biomarker is a disease feature that categorizes an individual as affected or unaffected or classifies them into subcategories based on the disease.A susceptibility or risk biomarker would be used to represent an individual with no apparent evidence of disease to reflect the risk of developing this kind of medical condition.A prognostic biomarker would be used in the baseline assessment of a patient with a confirmed diagnosis to predict the risk of occurrence of a clinical event.Predictive biomarkers would allow patients to be classified as candidates for medical intervention according to their probability of response.A biomarker of response can be measured in a patient who has already been exposed to pharmacological intervention to assess the biological response. They can be further classified into pharmacodynamics and surrogate endpoint biomarkers.

## 3. Fluid-Based Biomarkers in ALS: The Importance of the Study Sampling

Focusing on biological fluid-based biomarkers, other relevant points to address are the advantages and disadvantages of the different biofluids accessible in ALS patients.

Samples of cerebrospinal fluid (CSF): It has been widely used in research on neurodegenerative diseases in recent decades. Given its direct contact with the structures of the central nervous system, it represents the main reservoir of products derived from neuronal damage. Additionally, it is an ultrafiltrate of plasma with limited homeostatic mechanisms, and it is considered to be a comparatively low-complexity biofluid, simplifying the assay of low-concentration molecules. However, collection by lumbar puncture is invasive, can lead to complications, and access may be limited by the patient’s physical deterioration in the advanced stages of the disease. Hence, the use of CSF as a biofluid is limited when one is considering longitudinal measurements in the same patient, such as in the case of a potential response biomarker.

Blood and urine: They are alternatives that allow more suitable collection. It is relevant to note that these biofluids are not substitutes for CSF, but they can provide additional information. In the case of blood, it can reflect the pathological process of muscle destruction resulting from lower motor neuron degeneration more accurately than CSF can, which is an essential component of the neurodegeneration in ALS. Blood is a more complex fluid than CSF is, which makes it more difficult to detect and quantify biomarkers, especially protein biomarkers. Factors that compromise the utility of blood include the presence of high abundance proteins (e.g., albumin) that interfere with the detection of low concentration proteins, sequestration in different aggregates and the formation of immunocomplexes or changes in molecular processes due to the pathophysiology of the disease (e.g., clearance of misfolded proteins by chaperones). A more detailed description of these features is beyond the scope of this review, but we recommend the work of Sturmey et al. [13] for a more comprehensive discussion.

In contrast, the search for urinary biomarkers in ALS has scarcely been explored in smaller studies or those that have produced less consistent results. Biomarkers research using the saliva of ALS patients is also sparse, but it is a promising opportunity because of its ease of acquisition. Although it is beyond the purpose of this review on biological fluids, the exploration of the pathological signature of ALS in different tissues by several technologies, such as proteomic or metabolomic analyses, is a field of intriguing potential to try to complete the ALS puzzle.

## 4. Recent Developments of ALS Prognostic, Predictive and Response Biomarkers in Biological Fluids

The current statuses of a range of biomarkers in ALS are described by the pathophysiological pathway to which they belong. Due to their particular relevance, neurofilament proteins are covered in a separate section. Despite the limitations of the current evidence of biomarkers in urine compared to blood and CSF, we highlight some of the most recent and promising ones in separate sections.

### 4.1. Neurofilaments

Neurofilaments (Nfs) are neuron-specific cytoskeletal proteins belonging to the intermediate filament family. With a diameter of 8–10 nm, they are composed of heteropolymers of different subunits. They are classified according to their molecular mass in heavy (NfH), medium (NfM) and light (NfL) chains, and they are one of the fundamental scaffolding protein of axons [14,15]. In the case of NfM and NfH, they require post-translational modifications such as O-glycosylation or phosphorylation for proper stabilization and, consequently, to perform the correct function [14]. Since pioneering studies detecting elevated levels of Nfs (specifically of the NfL subunit) in the CSF of patients with Alzheimer’s disease and ALS [16], extensive research on Nfs has resulted in its emergence as an unspecific marker of acute and chronic axonal damage in the generic field of neurology, encompassing different types of insults to the central nervous system and, to a lesser extent, peripheral nervous system [17,18]. In the case of ALS, NfL and phosphorylated-NfH (p-NfH) subunits have been the focus of research efforts for their potential biomarker value. It is worth noting that these advances in the last decade have been driven by the development of new immunoassays, such as third generation (electrochemiluminescence) and, mostly, fourth generation ones (single-molecule array). These molecular techniques have allowed the highly sensitive and reliable determination of Nfs in blood [18].

Regarding the prognostic value of Nfs, accumulated evidence from several independent studies indicates a correlation with the rate of progression and other parameters of disease severity (such as a faster decline on the ALSFRS-R scale) and shorter lifespan. This prognostic utility appears to be valid in blood and CSF for both NfL [19,20,21,22] and p-NfH [19,21,23]. However, there are unresolved issues that remain to be clarified regarding the differences between these subunits. Recent studies have pointed to a certain superiority of NfL. A stronger correlation between serum and CSF levels [19] and its prognostic capacity is among these reasons [24,25]. More profound knowledge of the evolution of NfL levels throughout the natural history of the disease have made it: (1) a validated prognosis biomarker, (2) a robust option as a pharmacodynamic biomarker and (3) a risk biomarker of phenoconversion in some genetic forms of ALS [26]. While some studies on patients carrying pathogenic variants in different ALS-related genes only detected differences in Nfs levels when they were compared to familial controls in the early symptomatic phase [27], two other studies by the same authors found an elevation in serum NfL levels in patients with the pathogenic *SOD1* A4V2 variant up to 12 months before the onset of manifest symptoms [28] and 2 and 3.5 years in the case of *C9ORF72* and *FUS*, respectively [29]. The aggregate analysis of these investigations indicates variability in the exact timing of the onset at which Nfs begin to rise in the general population with genetic ALS. This is a likely reflection of the pathophysiological heterogeneity of the disease or the molecular mechanisms behind the genetic variations. However, its application in specific populations with more homogeneity can be of enormous utility for diagnosis in presymptomatic phases as a biomarker of the risk of phenoconversion. The importance of these tools will depend on the existence of effective therapies, but it can be exemplified by the ATLAS clinical trial with tofersen. In this study, Nfl elevation is used as an eligibility criterion as it is considered to be a biomarker of evidence of an active disease in *SOD1* variant carriers [30].

Regardless of doubts about the timing of Nfs level elevation, there is more agreement on the time course of NfL levels in the symptomatic phase of the disease, in which there is a rise in the initial symptom stages until a plateau is reached, and then NfL remains stable with disease evolution [19,24,31]. The concentration of NfL at which this plateau is established differs between patients, but higher concentrations of NfL confers the prognostic value linked to a more aggressive course [19,24,32]. The availability of an accessible and quantitative marker of neuro-axonal damage whose concentrations remain uniform in the same patient throughout the progression of the established disease is an optimal hypothetical starting scenario for the use of NfL as a biomarker of response in ALS. However, NfL cannot yet be accepted as a validated surrogate target using the previous experiences. In the phase 3 clinical trial with tofersen in *SOD1* patients [33], there was a significant reduction of plasma NfL levels without statistically significant changes in ALSFRS-R at week 28, the primary endpoint. Of note, nevertheless, a favorable trend was observed in secondary and exploratory clinical outcome measures that became more marked in the open-label extension study with longer term follow-up, particularly with earlier tofersen initiation [34]. In phase 2, AMX0035 with sodium phenylbutyrate–taurursodiol showed no differences in plasma p-NfH concentrations, but there were clinical differences in the change in the ALSFRS-R score. Nonetheless, the use of Nfs as a pharmacodynamic biomarker to signal the existence of biological activity of an experimental therapy may be of substantial utility, similar to clinical trials with other motor neuron diseases, such as spinal muscular atrophy [35]. The introduction of serum NfL as a validated pharmacodynamic biomarker in clinical trials has already been proposed [26], with an emphasis on one scenario: phase 2 clinical trials to accelerate the acquisition of results and the decision to advance to phase 3, where clinical efficacy must be demonstrated.

### 4.2. Neuroinflammation

Although it has not been fully disentangled, there is a robust body of evidence for the dysregulation of both the innate and adaptive immune systems as major players in the pathological process of ALS. The most accepted hypothesis affirms the existence of an initial phase of protective immune activation, promoting neuronal repair processes, and a pro-inflammatory and neurotoxic phase as the disease progresses [36]. Understanding the potential role of neuroinflammation as a cause or driver of neurodegeneration in ALS has encouraged scientific efforts to find immunological therapies to modify the natural history of the disease [37]. However, in addition, the possibility of identifying evidence of immune activation in biological fluids has broadened the field of biomarkers of neuroinflammation.

Numerous studies have examined the role of several **cytokines and inflammatory mediators** in ALS. The pooled results point to dysregulation of many of these cytokines measured in plasma, serum or CSF. As it was recently noted in a recent review article [38], the results have been mixed regarding prognostic capacity. These inconsistencies in predicting progression or survival may be due to different sample sizes, analysis technologies or differences in the disease stage or concomitant processes at the time of sample collection (note the mostly retrospective, not longitudinal, nature of these studies). Interleukin-6 (IL-6) is one of the most frequently evaluated cytokines, and it may illustrate this situation. The authors of a 2020 study in a population-based cohort of 79 ALS patients found that plasma IL-6 concentration correlated with the ALSFRS-R score, manual muscle testing and progression [39]. In contrast, the study by Devos et al. [40], which looked at a series of markers measured at four time points in a follow-up of up to 18 months in a larger cohort, did not find IL-6 among the predictors of progression on the ALSFRS-R scale. Other inflammatory parameters have also shown varied results. Serum **C-reactive protein (CRP)** has been related to the speed of disease progression in a cohort of 394 ALS patients [41] and to mortality risk in another cohort of almost 400 individuals [42]. However, these findings were not replicated in a 2017 population-based study in a German ALS registry [43]. **Chitinases** belong to a large family of hydrolases with potential relevance in different neurological diseases under the premise that they may function as a marker of glial activity. The elevation of the quantity of chitinases in the context of ALS patients was first described in a proteomic study [44], specifically Chitotriosidase (CHIT1), and successive studies evaluating the diagnostic and prognostic value of CHIT1, chitinase-3-like protein-3 1 (YKL-40) and chitinase -3-like protein-2 (YKL-39) have since been published. Although the evidence supporting the diagnostic utility of the CSF measurement of these chitinases is more consistent, some studies have found that they have a prognostic performance, correlating inversely with the rate of progression [45,46,47] and, in some cases, with survival [46,47]. Interestingly, the authors of a 2021 study [48] selected three neuroinflammatory markers in CSF: NfL, as a marker of neuronal damage, CHIT1 and CHI3L1 (YKL-40), as indicators of glial activity, and monocyte chemoattractant protein (MCP1), as a marker of peripheral immune activity. Univariate analysis confirmed the ability to predict the survival of each marker separately but, more importantly, the combination of NfL and CHI3L1 appeared to be a more robust predictor of survival. This underlines the importance of testing strategies that combine different immune pathways simultaneously.

One possibility proposed to overcome the limitations of raw measurement of inflammatory mediators is to assess the gene or protein expression levels (including surface markers) in immune cell subpopulations in peripheral blood. This approach has led to **immunophenotyping** studies such as the one published in 2017 by Gustafson et al. [49], in which patients could be classified into two immune profiles based on lymphocyte and monocyte phenotypes. These profiles were associated with different ages of onset and survival. Another prominent example investigating the value of studying specific immune populations is a study in a longitudinal cohort that identified a significant correlation of early changes in neutrophils and CD4 T cells with disease progression, as measured by changes in the ALSFRS-R score [50]. Overlapping with the above work, the study of **regulatory T cells (Treg)** has also shown their promising value as a prognostic biomarker. This cellular subtype has a relevant role in immune self-tolerance and performs has a suppressive effect on different elements of immune response. Various studies have suggested a neuroprotective effect of these cells and have reported that Treg cell dysregulation is related to the rate of disease progression [51,52,53].

Addressing the humoral immune response in ALS, the detection of **autoantibodies and immunocomplexes against Nfs** seems to be related to disease progression [54]. In addition to providing more insight into immune dysregulation in ALS, it may be a novel way to study biomarkers. The cost effectiveness of measuring more abundant antibodies rather than measuring the corresponding antigens could facilitate research.

### 4.3. Metabolism

Although there is considerable variability among the observations from numerous studies, a trend toward the up-regulation of certain serum and plasma metabolites in ALS can be identified and is consistent with the abnormalities in energy metabolism in the disease ranging hypermetabolism, altered glucose and lipid metabolism and mithocondrial dysfunction [55].

It is established that body mass index (BMI) is related to ALS risk and prognosis [3]. Many molecules related to **glucose and lipid metabolism** have been linked to ALS risk, but none of these have yet been validated as a fluid-based biomarker that can be paired with BMI. In a 20-year follow-up study of a Swedish cohort of more than 600,000 individuals, Mariosa et al. [56] found that there is an increased risk of developing ALS with an elevated low-density lipoprotein cholesterol (LDL-C), apolipoprotein B (apoB), and LDL-C/high-density lipoprotein cholesterol (HDL-C) and apoB/apoA-I ratios. When the prognostic potential of these lipids has been explored, the results have been conflicting. An association has been detected between a lower risk of death from ALS after diagnosis and an increase in the levels of these same lipids (along with total cholesterol) [57] and between decreased levels of total cholesterol, LDL-C and a lower LDL-C/HDL-C ratios and respiratory impairment [58,59]. However, in a study such as the one by Paganoni and colleagues with a cohort of over 400 patients from 3 clinical trial databases, they found no association between the LDL-C/HDL-C ratio and survival [60]. These mixed results were confirmed in a recent meta-analysis [61], which concluded that there was a lot of heterogeneity in the design and statistical methodology. The second part of this work was a population-based study that found that the prognostic utility was restricted to an increase in HDL-C, which was associated with a poorer survival rate. In conclusion, more impactful longitudinal studies are required to confirm whether there is a relationship between lipids and disease prognosis or whether lipid variations are a consequence of the disease’s progression. Finally, beyond generic biochemical measurement studies, there are emerging platforms for the analysis of lipid metabolites, such as lipidomics, whose application may provide new insights into ALS biomarkers. Several studies have mainly investigated the lipidomic signature of ALS patients in CSF samples, and to a lesser extent, in blood [62,63]. For instance, a recent work by Sol et al. found an association between specific lipid profiles and cases of rapid progression [62]. The same study also found relevant variations in the level of lipidome depending on the predominant location of disease onset. These results indicate a promising diagnostic and prognostic capability of lipid profiling through the development of these technologies.

Likewise, glucose metabolism is also impaired in ALS [64], and elevated serum glucose levels have been associated with an increased risk of the disease [56]. Although there is growing evidence that a higher metabolic level is associated with a faster decline and decreased survival [65,66], a few studies have identified a fluid-based biomarker with a prognostic value reflecting glycolytic up-regulation. Prior research identified blood hemoglobin A1c as an independent predictor of increased mortality [67], measured at the time of ALS diagnosis. However, the most noteworthy investigation is a retrospective 2020 study of a large cohort of ALS patients, evaluating the prognostic ability of different routine blood markers. The authors found that the baseline elevation of serum glucose at the time of diagnosis was associated with an increased risk of mortality, as well as its elevation over time [42].

**Altered iron metabolism** affects oxidative stress and ferroptosis pathways, and iron dyshomeostasis may have a prominent role in neurodegenerative processes such as ALS [68]. Several publications have studied a link between iron markers and diagnosis, progression and survival in ALS. These studies shown a trend towards increased ferritin and decreased transferrin levels in patients compared to those of the controls [69]. In terms of prognosis, the results have been slightly contradictory between studies, but with a tendency to point to ferritin as a potential prognostic biomarker, as endorsed in a meta-analysis published in 2021 that showed that serum ferritin level was negatively associated with the overall survival of ALS patients [70]. A recent novel approach has been the finding of a negative correlation between serum ferritin levels with disease duration and a positive correlation with the rate of disease progression [71]. The same study identified a significant increase in CSF ferritin levels in ALS patients, but this had no relation to the disease’s progression.

Another metabolite worth mentioning is **serum albumin**. The authors of a population-based study conducted in Italy in 2014 on a large cohort of over 700 ALS patients identified that serum albumin measurement was an independent predictor of survival (with better outcomes with higher levels) [72]. There was an absence of a correlation with body mass index, but a significant correlation was detected with inflammation-related parameters such as the erythrocyte sedimentation rate and leukocytes. These findings suggested that albumin alterations in ALS may be caused by the proinflammatory state. Subsequent studies have found more limited value as a single point-in-time measure [73], but more consistently, the longitudinal decrement appears to predict functional decline and survival [42,74].

### 4.4. Muscle Injury Biomarkers

An important area of interest has been the hypothesis that muscle damage by-products, such as plasma creatinine or creatine kinase, may reflect the degree of muscle denervation in ALS, and thus, be related to the aggressiveness and severity of the disease.

In the case of **plasma creatinine**, evidence of strong longitudinal correlations with muscle strength, ALSFRS-R and overall mortality were found in the analysis of 1200 patients from three different clinical trials cohorts in a study published in 2018 [75]. The authors themselves suggest that using plasma creatinine as a biomarker of response could reduce the sample size needed for 18 month trials by 21.5%. The potential of creatinine as a prognostic biomarker was reinforced in a meta-analysis and systematic review conducted by Lanznaster et al. in 2019 [76], although it pointed to a need for standardized criteria and a methodology to validate plasma creatinine as a clinical biomarker. Since then, new studies have strengthened the argument that the measurement of plasma creatinine, as well as other metabolites, at the time of diagnosis and their changes with temporal monitoring and disease progression have a strong prognostic potential [42,77,78].

Similarly, baseline **creatine kinase** levels and their evolution over time show potential association with clinical aggressiveness and survival. Higher levels at the time of diagnosis would be associated with a better survival rate [79,80,81,82] and faster declining levels with a more aggressive disease, as the analysis derived from the PRO-ACT database revealed [74]. Interestingly, some of these studies found correlations between the two markers, supporting that they express similar information about skeletal muscle health statuses [81,82].

Recently, two studies have suggested that **cardiac troponin T (cTnT)** is an easily accessible marker of interest in ALS [83,84]. Both studies found a diagnostic ability and a progressive elevation over the course of the disease, but the differed in prognostic accuracy. While in the study by Kläppe et al., it was not associated with lower survival in a multivariable survival model, in the work of Castro-Gomez and colleagues, it correlated with clinical severity, as measured by the ALSFRS-R score. Although the causes are unclear, the correlation with motor domains in ALSFRS-R, but not with bulbar symptoms points, neither with p-NfH levels in CSF and serum cardiac troponin I (cTnI), point to the origin of cTnT in skeletal muscle.

### 4.5. Non-Coding micro-RNA (miRNA)

The dysregulation of epigenetic mechanisms, particularly miRNAs, has been associated with a number of neurodegenerative diseases, including ALS [85]. Following these hypotheses, the investigation of miRNA profiling could be a promising approach in biomarker research.

The main obstacles to date are the lack of consistency and overlap in specific ALS-associated miRNAs between the different study groups and a lack of longitudinal studies. In addition, interest has focused on the characterization of the circulating miRNA profile in ALS [86,87] and its diagnostic ability compared with that of other diseases and in controls. However, their prognostic performance has been explored less often. In this regard, it is noteworthy that a recent study has detected plasma miR-181 as a robust prognostic biomarker [88]. Plasma miR-181 produced a two-fold increased risk of death in two independent cohorts of ALS patients. Furthermore, in this investigation, the miR-181 performance was found to be similar to that of NfL, but its combination synergistically increased the prognostic accuracy.

### 4.6. Specific Biomarkers in Genetic Subtypes of ALS

The development of gene silencing therapy applied in ALS has highlighted the leading exponents of antisense oligonucleotide therapies in several forms of genetic ALS. This includes: (1) the hexanucleotide repeat expansion in *C9ORF72* gene (C9-ALS) and (2) the pathogenic variants in the *SOD1* gene (SOD1-ALS), and (3) it has also driven the research of biomarkers associated with pathological products caused by the mutation.

In C9-ALS, it is possible to quantify a protein called **dipeptide repeat protein poly-GP** in CSF. This is a product derived from the abnormal processing of RNA, which is a consequence of genetic alteration [89]. The levels of these dipeptides are increased in the CSF of patients with C9-ALS or *C9ORF72*-associated frontotemporal dementia (C9-DFT), including preclinical stages [90], but they do not seem to correlate with the rate of progression [91]. However, the findings that the levels of dipeptide repeat appear to be stable over time and decrease after ASO therapy in cellular and animal models of C9-ALS [92] have made it a serious candidate for a pharmacodynamic biomarker in C9-ALS. It was incorporated into the clinical trial with ASO BIIB078 sponsored by Biogen in patients with C9-ALS. However, this trial has been discontinued for failing to show a clinical benefit despite achieving a decrease in the level of this biomarker [93].

Similarly, **SOD1 protein levels** in CSF have also been analyzed in the development trajectory of ASO therapies, with SOD1-ALS as a pharmacodynamic biomarker. Successive clinical trials with tofersen in patients with SOD1-ALS found a significant decrease in SOD1 peptide levels together with those of Nfl, but the phase 3 clinical trial did not meet the primary endpoint [33], as was discussed in the Neurofilament Section.

### 4.7. Urinay Biomarkers

The simplicity of their collection and the possibility of providing additional information to complement blood and CSF have encouraged research into urine-based biomarkers in recent years. Until then, it had been scarce and focused on trace elements with contradictory results [94,95]. We will highlight two biomarkers that show promising results and constitute a novel approach to studying neuroinflammation and neuronal damage mechanisms in ALS.

Neurotrophins are a family of growth factors that stimulate neuronal differentiation and survival. The urinary concentrations of the **extracellular domain of Neurotrophin receptor p75 (p75NTR^ECD^ or p75^ECD^)** represent a generic indicator of motor neuron degeneration. This evidence is based on: (1) the expression of p75 in motor neuron rodent models during embryonic development [96], (2) its re-expression after peripheral nerve injury [97], as well as in postmortem nerve tissues of humans with ALS [98] and (3) the urinary elevation of p75^ECD^ levels in *SOD1^G93A^* mouse models and ALS patients [99]. Subsequent studies [99,100,101] and a recent meta-analysis [102] have expanded the potential of this biomarker in ALS. The results showed that p75^ECD^ could be detected before symptoms onset in *SOD1^G93A^* mice. They also detected that its levels in ALS patients do not remain stably elevated, but increase with disease progression (average rate of 0.19 ng/mg creatinine/month) and correlate with the ALSFRS-R score. In addition, it may add prognostic value at the baseline to the usual clinical parameters, such as bulbar onset or ALSFRS-R slope [101]. These characteristics make p75^ECD^ a promising biomarker with prognostic significance, and also, as a biomarker of phenoconversion and disease progression, raising the possibility of its use as a pharmacodynamic biomarker.

**Neopterin** is a molecule peripherally released by mononuclear cells such as microglia in the CNS and dendritic cells and macrophages [103,104]. Its release is induced by the stimulation of interferon γ (IFN-γ) and tumor necrosis factor α (TNF-α), in turn generated by Th1 and Th17 lymphocytes. Therefore, the measurement of neopterin has been used to assess the shift to the pro-inflammatory state [105,106], in which these cells predominate. This is an event that is part of immune dysregulation, which occurs in ALS [36]. The potential of urinary neopterin as a biomarker in ALS was raised in a previous study [106], but its relevance has recently emerged in an observational study [107]. This work observed a progressive elevation of Neopterin as the disease advances, introducing possibilities for its use as a biomarker of progression and pharmacodynamics. The authors of the same study explored the prognostic potential of urinary neopterin and urinary p75^ECD^, but found no added value in the multivariate analysis to that provided by the ALSFRS-R progression rate.

These two novel molecules represent a complementary approach to blood and CSF in the study of markers of immune dysregulation in neurodegenerative diseases. They also propose that the study of biomarkers in the less complex matrix that is urine can be very valuable in diseases with a complex immune background such as ALS.

## 5. New Horizons in ALS Biomarker Research: An Insight into the Pathology Associated with TDP-43

In the previous sections, we have tried to report the efforts performed by the scientific community to find prognostic and response biomarkers, addressing the best-known pathological mechanisms in ALS. A good example is the recent findings on the pathogenic mechanisms underlying TDP-43 dysfunction. The mis-localization from the nucleus to the cytosol of this RNA-binding protein is the hallmark pathological feature in the majority of ALS cases and FTD. One of its altered functions is the repression of the so-called cryptic exons during RNA splicing. Recent studies [108,109] have found that one of the genes that undergoes increased random inclusion of cryptic exons, and this leads to decreased expression of its protein is unc-13 homolog A (*UNC13A*). These findings support the hypothesis that the failure of TDP-43 control function in RNA processing and the consequent generation of abnormal proteins have a relevant toxic effect on the pathophysiology of the disease. The probability of the inclusion of cryptic exons in the *UNC13A* gene is increased in the presence of two specific single-nucleotide polymorphisms (SNP), which had previously been associated with an elevated risk of ALS in genome-wide association studies. One of them (SNP rs12608932) was associated even with an earlier onset and shorter survival time [110,111,112]. In addition, *UNC13A* genotyping has been proposed as a potential example of a genetically tailored treatment following the results of the post hoc analysis performed in three clinical trials with lithium [113]. These analyses showed a different effect of lithium in homozygous carriers of the C allele at SNP rs12608932 in *UNC13A*, leading to a 40.1% improvement in the probability of survival at 12 months. Currently, a multicenter clinical trial with lithium is being conducted in patients homozygous for this allele in *UNC13A* [114], thus constituting *UNC13A* genotyping as a true predictive biomarker.

The role of TDP-43 in RNA splicing affecting several proteins potentially involved in the pathogenesis of ALS has been the subject of a previous investigation. A very appropriate example is Stathmin-2 (STMN2). STMN2 is a protein involved in neuronal repair and axonal stabilization [115], and molecules are currently being developed to restore its expression. However, a link between TDP-43 dysfunction and genetic risk factors has been established for the first time with *UNC13A* and introduces the possibility of seeking a new generation of biomarkers related to cryptic exons in TDP-43 proteinopathies.

Linked to the above, the increasing knowledge about the genetic aspects of ALS may further broaden the horizons of biomarker research. A concrete fact that can illustrate this statement is the increased knowledge about several genes that work as a risk factor or phenotype modifier. The genotyping of ALS patients, including these genes, may provide relevant prognostic information. In addition to the aforementioned one, *UNC13A*, another example is the ataxin-2 (*ATXN2*) gene. Intermediate-length polyglutamine (polyQ) expansions in the ataxin-2 (*ATXN2*) gene are associated with an increased risk of ALS [116]. The reduction of ataxin-2 appears to ameliorate TDP-43 pathology and may be a therapeutic tool to be developed in future clinical trials [117].

## 6. Conclusions

In this review, we have tried to provide an overview of the fluid-based biomarkers that currently are supported by the most evidence as prognostic or response biomarkers (Table 2 summarizes the most prominent prognostic biomarkers discussed in this review). Although, undeniably, there are still many limitations, including the absence of an effective therapy, as in the last decade, we have witnessed an expansion of our knowledge of ALS. This path has allowed us to discover numerous biomarker candidates in already known aspects of the disease, but also to explore the possibilities of the new insights that were being incorporated (such as genetic or proteomic aspects). As a result, we are currently facing the existence of a multiplicity of potential biomarkers that have not yet “leaped” routine clinical incorporation or clinical trial research. Based on the existing evidence and highlighted in the corresponding section, neurofilaments are the biomarker that is closest to progressing to this level. Although the analysis and description of the difficulties and next steps in the discovery and validation of biomarkers in ALS are not the focus of the present review (we recommend the work of Benatar et al. [118]), we can outline some key areas for progress in the field of ALS biomarkers on which there is agreement.

On the one hand, although emerging areas of disease research may serve to discover new biomarkers, a concurrent effort is required to advance the validation of potential parameters already known so that they can be ruled out or pushed forward as useful biomarkers. This validation is a highly complex task, but it requires prospective studies with standardized operational procedures and the participation and collaboration of different medical centers. Even though the use of databases and open-access initiatives have brought relevant advances, some of which are mentioned in this review, multicenter studies are of great importance due to the need to validate the collection and measurement methods. This is a key concept shared by the scientific community in ALS. The clinical validation of biomarkers should be performed in prospective cohorts involving large numbers of patients who are systematically evaluated at multiple centers. This should include uniform procedures for phenotyping, as well as for the collection, processing and analysis of biological samples. Investigators should address measures to minimize sources of variability (intra- and inter-subject, intra- and inter-assessment and interlaboratory ones). This aspect is probably one of the fundamental reasons for the inconsistencies found in the literature in the different studies, and many of them have been mentioned in this review. Indeed, multisite clinical trials are a suitable scenario to fulfil these criteria. This is one of the reasons for the importance of collaborative efforts between the different actors involved in the disease, including the pharmaceutical industry.

Finally, advances towards revealing the complexity of the underlying mechanisms in ALS provide another relevant insight for the search for biomarkers; likely, a single marker will not be able to capture the complex dynamics of the disease. Therefore, there is a growing interest in combining injury indicators in different tissues, which could multiply their benefit and provide a more global vision of the neurodegeneration process in any given patient [119].

## Figures and Tables

**Table 1 cells-12-01180-t001:** Types of biomarkers in the field of Amyotrophic Lateral Sclerosis research.

Biomarker Category	Time Point of Measurement	Description	Illustrative Example
Diagnostic Biomarker	One-off measurement (during the diagnostic study process)	A disease characteristic that classifies an individual according to the existence or lack of a particular condition. It should be a useful measure for the clinician to make diagnostic decisions in individuals with subtle or inconclusive impairment.	Needle EMG findings demonstrating subclinical lower motor neuron pathology (active and chronic denervation) are included in the commonly used diagnostic criteria.
Susceptibility/Risk Biomarker	One-off measurement	A marker that reflects the likelihood of a specific medical condition or clinical event (progression) in an individual with no evidence of that particular disease.	The elevation of NfL in patients carrying pathogenic variants in certain ALS-associated genes could identify patients at risk of disease onset and to select them for early targeted therapies, such as *SOD1* variants in the ATLAS clinical trial with Tofersen.
Prognostic biomarkers	One-off measurement (baseline)	A baseline measure that allows a patient to be categorized into different risk levels based on the probability of occurrence of a clinical event, which could be for example rate of progression, death, etc.	Selected pathogenic variants in the *SOD1* (A4V) gene or *UCL13A* SNPs have been associated with a poor prognosis.
Predictive biomarker	One-off measurement (baseline or stratification for trial design)	A marker that, when performed at the time of a patient’s baseline assessment, allows the estimation of the likelihood of benefiting from a medical intervention.	*UNC13A* genotype may have a positive modifying effect on ALS patients in post hoc analysis of lithium clinical trials.
Response biomarker	Longitudinal measurements	Pharmacodynamic biomarkers: in a patient who has already received a pharmacological intervention, it may be indicative of the biological activity of the drug, irrespective of clinical efficacy. Surrogate endpoint biomarkers: indirect indicator of a study outcome, allowing substitution for a direct measure.	Tracking of the protein encoded by ALS-associated genes may be a valuable biomarker in both preclinical studies and human trials, such as dipeptide repeat protein poly-GP in *C9orf72-*ALS.

EMG: electromyography; NfL: light chain neurofilaments; *SOD1*: superoxide dismutase 1. SNP: single-nucleotide polymorphisms.

**Table 2 cells-12-01180-t002:** Summary of prognostic biomarkers outlined in this review in each pathophysiological section.

Pathophysiological Pathway	Biomarker	Biological Fluid	Prognostic Performance Commentary	Key References
Neuro-axonal damage	NfL	Blood and CSF	- Higher plasma Nfl concentrations are associated with more aggressive evolution, specifically the rate of progression of functional decline as measured by the ALSFRS-R and shorter survival. - Stronger correlation between serum and CSF Nfl.	Thompson et al., 2022 [24] Benatar et al., 2020 [19] Benatar et al., 2020 [19]
	pNfH	Blood and CSF	- Higher serum pNfH concentration is a negative independent prognostic factor for survival. - Levels pNFH in serum and CSF were positively correlated with ALSFRS-R slope.	Falzone et al., 2020 [23] Shi et al., 2022 [21]
Neuroinflammation	Chitinases	CSF	- Elevated CSF CHIT1was associated with increased hazard of death, and higher CSF levels of CHIT1 and CHI3L2 were correlated with higher ALSFRS-R slope. - CHI3LI (YKL-40) combined with Nfl was inversely associated with survival.	Thompson et al., 2019 [46] Masrori et al., 2021 [48]
	Antibodies and immunocomplexes against Nfs	Blood	- Lower NfH antibodies levels were associated with longer survival. - Increasing levels of Nfs antibodies and immune complexes between time points were observed in faster progressing ALS.	Puentes et al., 2021 [54]
	Immunophenotyping	Blood	- ALS patients clustered into an abnormal leukocyte profile survived longer than those who clustered similarly with healthy volunteers did. - Neutrophil and CD4 T-cell numbers were correlated with rapid disease progression.	Gustafson et al., 2017 [49] Murdock et al., 2017 [50]
	Tregs	Blood	- Inverse correlation between total Treg levels and rate of disease progression (ALSFRS-R slope).	Sheean et al., 2017 [53]
	Neopterine and p75^ECD^	Urine	- Urinary levels increase with disease progression and can be used as a pharmacodynamic biomarker and can be used as a pharmacodynamic biomarker.	Shepheard et al., 2017 [101] Shepheard et al., 2022 [107]
Metabolism	Lipid metabolism	Blood	- Increased HDL-cholesterol was associated with poorer survival rate.	Janse van Mantgem et al., 2022 [61]
	Glucose metabolism	Blood	- Median level of serum glucose at baseline and SD increase in longitudinal measurements were associated with a higher mortality risk.	Sun et al., 2020 [42]
	Iron metabolism	Blood	- Serum ferritin level was negatively associated with the overall survival. - Serum ferritin levels were negatively correlated with the disease duration and were positively correlated with the ALSFRS-R slope.	Cheng et al., 2021 [70] Paydarnia et al., 2021 [71]
	Albumin	Blood	- Albumin decreased in fast progressing patients and increased in slow progressing patients (ALSFRS-R decline).	Ong et al., 2017 [74]
Skeletal muscle damage	Plasma creatinine	Blood	- - Patients with a high rate of decline on ALSFRS-R (more prominent in the motor items) had a high rate of decline in plasma creatinine. - - There was an approximate linear relationship between the mean level of plasma creatinine and the mean muscle strength. - - Low baseline plasma creatinine levels and their longitudinal decline are associated with increased risk of mortality.	Van Eijk et al., 2018 [75]
	Creatine Kinase	Blood	- Slow progressors (ALSFRS-R slope) were associated with higher CK levels at baseline. - CK level decreased more sharply in the fast-progressing patients (ALSFRS-R decline) compared to slow-progressing patients.	Ceccanti et al., 2020 [80] Ong et al., 2017 [74]
RNA metabolism	miRNA-181	Blood	- High levels of miR-181 predicts a more than 2-fold risk of death.	Magen et al., 2021 [88]

NfL: neurofilament light chain. pNfH: phosphorylated neurofilament heavy chain. CSF: cerebrospinal fluid. ALSFRS-R: Amyotrophic Lateral Sclerosis Functional Rating Scale Revised. Nfs: neurofilament proteins Treg: regulatory T cells. p75^ECD^: extracellular domain of Neurotrophin receptor p75. HDL-cholesterol: high-density lipoprotein cholesterol. CK: creatine kinase. miRNA: Non-coding micro-RNA.

## Data Availability

No new data were created or analyzed in this study. Data sharing is not applicable to this article.

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
