# Peer review of "Biofluid Biomarkers in the Prognosis of Amyotrophic Lateral Sclerosis: Recent Developments and Therapeutic Applications"

_cells, 2023, doi:10.3390/cells12081180_

Round 1

Reviewer 1 Report

I enjoyed reading this review.  Still, there are a very few comments and changes this reviewer would suggest.

 In general, the separation of the different sections is not well structure and it makes it difficult to follow the rationale of the main messages. The introduction should be clearer on what is coming later, and there should be a better connection between the different sections. It reads as if there are separated parts put together, with no particular order. And if there is, it should be more clearly specified. A few examples are:

i/ Section 2, to separate the types of biomarkers, tables and description, from the advantages and disadvantages of fluid biomarkers. It doesn’t have a nice reading flow.

(page 3, starting at line 93).

Ii/ From section 2, goes to section 3 (page 4, line 123), which talks about a particular biomarker (neurofilaments). Again, from one thing jumps to another with no particular rationale of why.

Iii/ Some sections are one biomarker (such as neurofilament), but the others are a group of biomarkers,  related to the pathways or pathology they represent. Then again, the section 9 is urinary biomarkers, but the category doesn’t follow the previous rationale.

For all these, this reviewer suggests to re-structure the whole manuscript and have a more organised and categorized flow in presenting the ideas and concepts.

Other considerations:

- within the metabolism group, the authors mention lipid metabolism, but then they only referred the studies conducted in blood samples by general biochemistry (classical measurements of clin biochemistry analysis). It should be noted that new technologies are emerging and now there are more sophisticated platforms for the analysis of lipid metabolites, such us lipidomics. There are a few serum lipid profiles described for ALS patients that should be noted.

- A more pronounced comment and debate would be good, in relation to the huge variability and heterogeneity in the results amongst the different studies, for most biomarkers. The authors made some comments in some individual sections, but this reviewer considers that it should have a separated discussion, as it is a generalized problem. This is one of the biggest problems nowadays, the inconsistencies found in the literature of the different studies results. The authors might want to bring up this important point, and even mention which studies might have more credibility according to the most adequate study design.

- The genetic biomarkers are not considered here in a separated section. These are also biomarkers to take into consideration, at least they should be mentioned. The section 8 talks about specific proteins that are measured for two particular genetic forms of ALS, but those are not genetic markers per se. A comment or paragraph about genetic analysis should be not forgotten here, like it or not, as there is extensive research dedicated to that nowadays that cannot be ignored.

Author Response

ALS Unit

Neurology Department

Hospital Universitari Vall Hebron-UAB

Passeig Vall d`Hebron 119

080035 Barcelona

SPAIN

April 14, 2023

Ms. Harper Shang

Assistant Editor

CELLS

Dear  Ms Shang

Please find attached the revised version of the manuscript entitled “Biofluid biomarkers in the prognosis of amyotrophic lateral sclerosis: recent developments and therapeutic applications” (ID: cells-2300354) for consideration for publication in your Journal CELLS as an REview.

As regards the reviewers’ comments:

REVIEWER 1

I enjoyed reading this review.  Still, there are a very few comments and changes this reviewer would suggest.

 In general, the separation of the different sections is not well structure and it makes it difficult to follow the rationale of the main messages. The introduction should be clearer on what is coming later, and there should be a better connection between the different sections. It reads as if there are separated parts put together, with no particular order. And if there is, it should be more clearly specified. A few examples are:

  • i/ Section 2, to separate the types of biomarkers, tables and description, from the advantages and disadvantages of fluid biomarkers. It doesn’t have a nice reading flow (page 3, starting at line 93).
  • Ii/ From section 2, goes to section 3 (page 4, line 123), which talks about a particular biomarker (neurofilaments). Again, from one thing jumps to another with no particular rationale of why.
  • Iii/ Some sections are one biomarker (such as neurofilament), but the others are a group of biomarkers, related to the pathways or pathology they represent. Then again, the section 9 is urinary biomarkers, but the category doesn’t follow the previous rationale.

For all these, this reviewer suggests to re-structure the whole manuscript and have a more organised and categorized flow in presenting the ideas and concepts.

We have reorganized the sections into which the manuscript is organized, following some of the indications raised by the reviewer. The objective has been to improve the reading and comprehension of the ideas presented, as suggested by the reviewer.

In addition, we have added information at the end of the introduction section to make the organization of the subsequent sections more understandable.

Other considerations:

- within the metabolism group, the authors mention lipid metabolism, but then they only referred the studies conducted in blood samples by general biochemistry (classical measurements of clin biochemistry analysis). It should be noted that new technologies are emerging and now there are more sophisticated platforms for the analysis of lipid metabolites, such us lipidomics. There are a few serum lipid profiles described for ALS patients that should be noted.

We agree with the reviewer on the importance of these emerging platforms and their promising application in the field of ALS biomarkers and appreciate the suggestion. We have incorporated a reference to this at the end of the paragraph on lipid metabolism markers and have made reference to two recent articles that we considered of interest.

- A more pronounced comment and debate would be good, in relation to the huge variability and heterogeneity in the results amongst the different studies, for most biomarkers. The authors made some comments in some individual sections, but this reviewer considers that it should have a separated discussion, as it is a generalized problem. This is one of the biggest problems nowadays, the inconsistencies found in the literature of the different studies results. The authors might want to bring up this important point, and even mention which studies might have more credibility according to the most adequate study design.

We fully agree with the reviewer that there is currently a whole plethora of candidate biomarkers but significant and extensive inconsistencies between the different studies. We believe that this point should be emphasized in a review such as this one and therefore decided to devote the central part of the conclusions section to this point. We have decided to extend the discussion and debate in this section following the reviewer's indications. Likewise, we have highlighted the studies and the approach that we consider and is considered by most authors to be key in the validation of biomarkers. This is the need for prospective multicenter longitudinal studies that share harmonized clinical and paraclinical procedures.

- The genetic biomarkers are not considered here in a separated section. These are also biomarkers to take into consideration, at least they should be mentioned. The section 8 talks about specific proteins that are measured for two particular genetic forms of ALS, but those are not genetic markers per se. A comment or paragraph about genetic analysis should be not forgotten here, like it or not, as there is extensive research dedicated to that nowadays that cannot be ignored.

We fully agree with the reviewer about the importance of genotyping in ALS patients and the biomarker and therapeutic research possibilities it allows. We have added a paragraph about it and completed it with annotations about therapeutic research on genes such as STMN2 and Ataxin-2, as also indicated by another reviewer.

We have included these annotations in the last section and we have reformulated it with the idea of giving a vision of the new genetic aspects and of the deepening in the knowledge of the TDP-43 pathology. In doing so, we believe that we have also followed the reviewer's indication to provide the article with greater internal coherence.

We thank the reviewers for the insightful comments, which have improved the manuscript. We hope we have addressed the reviewers’ concerns.

Thank you in advance for reconsidering our manuscript.

Sincerely,

Daniel Sanchez Tejerina a

              Department of Neurology

              Hospital Universitari Vall d’Hebron

              Passeig Vall d’Hebron 119-135

              08035 Barcelona, SPAIN.

Reviewer 2 Report

Dear Authors, 

I find the article very interesting in terms of the topic. It is very well-written and structured. It collects all the information in a systematic and up-to-date way. I have no comments because it seems to me that as it is, it is ready to be published. The only thing I noticed in table 1 is that in the "illustrative example" sections "Predictive biomarker" and "Response biomarker" two sentence dot endings are missing. Congratulations on your work. 

Best regards, 

Author Response

ALS Unit

Neurology Department

Hospital Universitari Vall Hebron-UAB

Passeig Vall d`Hebron 119

080035 Barcelona

SPAIN

April 14, 2023

Ms. Harper Shang

Assistant Editor

CELLS

Dear  Ms Shang

Please find attached the revised version of the manuscript entitled “Biofluid biomarkers in the prognosis of amyotrophic lateral sclerosis: recent developments and therapeutic applications” (ID: cells-2300354) for consideration for publication in your Journal CELLS as an REview.

As regards the reviewers’ comments:

REVIEWER 2

I find the article very interesting in terms of the topic. It is very well-written and structured. It collects all the information in a systematic and up-to-date way. I have no comments because it seems to me that as it is, it is ready to be published. The only thing I noticed in table 1 is that in the "illustrative example" sections "Predictive biomarker" and "Response biomarker" two sentence dot endings are missing. Congratulations on your work. 

We have made the corrections indicated in the text of table 1.

We thank the reviewers for the insightful comments, which have improved the manuscript. We hope we have addressed the reviewers’ concerns.

Thank you in advance for reconsidering our manuscript.

Sincerely,

Daniel Sanchez Tejerina and Jose Manuel Vidal Taboada

              Department of Neurology

              Hospital Universitari Vall d’Hebron

              Passeig Vall d’Hebron 119-135

              08035 Barcelona, SPAIN.

Reviewer 3 Report

Sanchez-Terejina et al. present a nice basic review on the subject of biomarkers for ALS.  The comprehensive nature of this review begins with a definition and then the pros and cons before discussing several examples.  The flow is good and the table gives a nice summary of the text helping the reader.  This is a complex subject, but readers will be interested in current status.

Some questions. 

1.  The article is focused on fluid biomarkers, but the first biomarker example in the table in EMG.  The authors may need to clarify that biomarkers can come in different forms.

2. In the sample section, there is mention of CSF, blood, and urine.  No mention of skin, breath, or other tissue. Noting it is likely beyond the focus, informing readers of these tissues on the horizon may be of note.

3. In the discussion of neurofilaments, there is no discussion of timing of change vs. clinic change.  The Toferson OLE data is not discussed.

4. No discussion of Stathman, Ataxin-2, and UNC13A as modifiers. All with clinic trials to start in near future.

Author Response

ALS Unit

Neurology Department

Hospital Universitari Vall Hebron-UAB

Passeig Vall d`Hebron 119

080035 Barcelona

SPAIN

April 14, 2023

Ms. Harper Shang

Assistant Editor

CELLS

Dear  Ms Shang

Please find attached the revised version of the manuscript entitled “Biofluid biomarkers in the prognosis of amyotrophic lateral sclerosis: recent developments and therapeutic applications” (ID: cells-2300354) for consideration for publication in your Journal CELLS as an REview.

As regards the reviewers’ comments:

REVIEWER 3

Some questions.

  1. The article is focused on fluid biomarkers, but the first biomarker example in the table in EMG. The authors may need to clarify that biomarkers can come in different forms.

Table 1 refers to the different generic categories of biomarkers following the terminology applied by the BEST initiative. Although our review is focused on fluid biomarkers, Table 1 is included in the section aimed at explaining the theoretical framework of biomarker work in the field of ALS. We understand the misunderstanding pointed out by the reviewer and have made an effort to structure the introduction and first sections in a more orderly and comprehensive manner before addressing the review of fluid biomarkers per se.

  1. In the sample section, there is mention of CSF, blood, and urine. No mention of skin, breath, or other tissue. Noting it is likely beyond the focus, informing readers of these tissues on the horizon may be of note.

As the reviewer indicates, research on biomarkers in tissues such as skin or muscle has unfortunately been left out of our scope for the present paper due to the need to focus on biomarkers in biological fluids. However, we agree on the relevance of this field of research and have considered including a sentence about it at the end of the section on the different biomarker samples in ALS.

  1. In the discussion of neurofilaments, there is no discussion of timing of change vs. clinic change. The Toferson OLE data is not discussed.

We consider that the text discusses the timing of neurofilament elevation and clinical changes (lines 150 to 170 in the version prior to this correction). The doubts about the exact time of elevation in the prodromal or presymptomatic phases of the disease and the greater consensus of its elevation in the symptomatic phases are mentioned. We may have misunderstood the reviewer's question. If you continue to feel that this is an important point for improving the text, we would ask you to let us know.

On the other hand, we have added a commentary on the results of the OLE study of the VALOR clinical trial.

  1. No discussion of Stathman, Ataxin-2, and UNC13A as modifiers. All with clinic trials to start in near future.

The role of UNCL13A in the pathogenesis of ALS and data on its role as a risk factor and phenotype modifier are mentioned in the "new horizons in ALS biomarker research" section. If the reviewer considers that a more detailed explanation is required, please let us know.

We have added a brief reference to the future role of STMN2 and ATXN2. We have included this annotations in the section about UNC13A and we have reformulated the section as new horizons in ALS biomarker research linked to a better understanding of TDP-43 pathology.

We believe that this organization provides an overview of the ongoing research linked to TFP-43 proteinopathies and that it may be of interest in this review, as the reviewer has pointed out.

We thank the reviewers for the insightful comments, which have improved the manuscript. We hope we have addressed the reviewers’ concerns.

Thank you in advance for reconsidering our manuscript.

Sincerely,

Daniel Sanchez Tejerina and Jose Manuel Vidal Taboada

              Department of Neurology

              Hospital Universitari Vall d’Hebron

              Passeig Vall d’Hebron 119-135

              08035 Barcelona, SPAIN.
